# Human and mouse activin genes: Divergent expression of activin A protein variants and identification of a novel heparan sulfate-binding domain in activin B

**Paul C. Billings** [ID] *, **Candice Bizzaro**◉, **Evan Yang**¤◉, **Juliet Chung**, **Christina Mundy**,
**Maurizio Pacifici**

Translational Research Program in Pediatric Orthopaedics, Division of Orthopaedic Surgery, The Children's
Hospital of Philadelphia, Philadelphia, PA, United States of America

◉ These authors contributed equally to this work.
¤ Current address: Graduate Program in Biological and Biomedical Sciences (BBS), Harvard Medical
School, Boston, MA, United States of America
* BillingsP@email.chop.edu

journal.pone.0229254

STATES

**Data Availability Statement:** All relevant data are
within the paper and its Supporting Information
files.

## Abstract

Activins are members of the transforming growth factor-β (TGF-β) superfamily of signaling
proteins and were originally identified as components of follicular fluid. The proteins are now
known to play critical roles in numerous normal and pathological processes and conditions,
but less is clear about the relationships between their gene organization and protein variant
expression and structure. The four human and mouse activin (Act) genes, termed *INHβA*,
*INHβB*, *INHβC* and *INHβE*, differ in exon numbers. Human *INHβA* is the most complex with
7 exons and elicits production of three Act A variants (Act A X1, X2 and X3) differing in their
pro-region, as we showed previously. Here we further analyzed the mouse *INHβA* gene and
found that its 4 exons encode for a single open reading frame (mouse Act A), corresponding
to the shortest human Act A X3 variant. Activins are synthesized and secreted as large com-
plexes made of a long pro-region and a short mature C- terminal ligand and are known to
interact with the heparan sulfate (HS) chains of cell surface and matrix proteoglycans.
Human Act A X1 and X2 variants do have a HS-binding domain (HBD) with Cardin/Wein-
traub traits in their pro-region, while the X3 variant does not as shown previously. We found
that the mouse Act A lacks a HBD as well. However, we identified a typical HBD in the pro-
region of both mouse and human Act B, and synthetic peptides containing that domain inter-
acted with immobilized HS and cell surface with nanomolar affinity. In sum, human and
mouse Act A genes elicit expression of different variant sets, while there is concordance in
Act B protein expression, reflecting possible evolutionary diversity in function of, and
responses to, these signaling proteins in the two species.

**Funding:** This work was supported by grants RO1AR061758 and RO1AR071946 from the Institute of Arthritis, Musculoskeletal and Skin Disease (NIAMS) at the National Institutes of Health to MP. The funders had no role in study design, data collection and analysis, decision to publish, or preparation of the manuscript.

**Competing interests:** The authors have declared that no competing interests exist.

**Abbreviations:** Act A, activin A; Act B, activin B; BMP, bone morphogenetic protein; CW, Cardin-Weintraub motif; ECM, extracellular matrix; FOP, Fibrodysplasia Ossificans Progressiva; HS, heparan sulfate; HBC, heparin binding complex; HBD, HS/heparin binding domain; HME, Hereditary Multiple Exostoses; TGF-β, transforming growth factor β..

## Introduction

Activins were originally described in the early 1900's as factors in follicular fluid regulating the release of gonadotropins from pituitary cells, but their identity remained obscure [1]. In 1986, follicle stimulating hormone (FSH)-releasing factor was identified in porcine ovarian follicular fluid based on its ability to induce the synthesis and release of FSH from anterior pituitary cells and was later named activin [2]. Since then, it has become apparent that in addition to their function in reproduction, activins are multifunctional cytokines, are expressed by different cell types and tissues, and play critical roles in a wide range of physiological processes, including growth and development, apoptosis, iron hemostasis, wound repair and pathological conditions including inflammation, liver disease, heterotopic ossification and cancer [3–5].

Activins are members of the transforming growth factor-beta (TGF-β) superfamily of signaling proteins that comprises over 40 members and includes TGF-βs, bone morphogenetic proteins (BMPs) and growth and differentiation factors (GDFs) [6]. All members are synthesized as large precursor pre-proproteins that are cleaved by furin/proprotein convertases intracellularly, resulting in a long N-terminal pro-region and a C-terminal active mature ligand [7, 8]. The mature ligands are dimers, with each monomer containing nine highly conserved Cys residues of which six participate in the formation of an intramolecular Cys-knot [9], stabilizing the dimers.

Activins exert their biological activity by interacting with cell surface tetrameric complexes consisting of 2 type I receptors (usually ALK4 or ALK7) and 2 type II receptors (usually ACVR2A or ACVR2B) [6]. As currently understood, the mature activin dimers initially bind to type II receptors with high affinity, and this results in recruitment of type I receptors and phosphorylation by the constitutively active type II kinase which activates the serine/thr kinase on the type I receptor [10]. The activated receptor complex subsequently phosphorylates SMAD2/3 proteins that combine with SMAD4 and translocate to the nucleus where they activate down-stream target genes.

Heparan sulfate proteoglycans (HSPGs) are important components of the cell surface and extracellular matrix and consist of a core protein with covalently attached heparan sulfate (HS) chains [11]. A major function of HSPGs is to selectively interact with cytokines and signaling proteins, thereby regulating protein distribution, bio-availability, activity and turnover [12–14]. The HSPGs can also serve as co-receptors as in the case of FGFs. Protein-HS interactions are mediated by short HS-binding domains (HBDs) that contain specific clusters of basic amino acids (Lys and Arg) with an overall net positive charge and a consensus Cardin–Weintraub (CW) motif arrangement [15]. Interestingly, alterations in HS biosynthesis can have severe physiological consequences as observed in pediatric patients with Hereditary Multiple Exostoses (HME), a congenital condition characterized by osteochondromas forming next to the growth plates (also known as Multiple Osteochondromas or MO) [16]. Most HME/MO patients carry a loss-of-function mutation in exostosin-1 (*EXT1*) or exostosin-2 (*EXT2*) genes encoding Golgi-resident glycosyltransferases responsible for HS biosynthesis and are thus HS-deficient [11].

Activins are synthesized and secreted as homo- or hetero-dimers with each monomer composed of the pro-region non-covalently associated with the C-terminal mature ligand. Following secretion, these complexes are thought to associate with HSPGs in the extracellular matrix or cell surface [17, 18]. Previously, we identified a classic HBD in the pro-region of human Act A variants X1 and X2, but significantly, a similar domain was absent in the shorter X3 variant [19]. This structural diversity and consequent differential HS binding could affect overall Act A variant distribution, availability and activity in human target tissues and organs. In the present report, we have extended our analyses to the mouse Act genes and protein characteristics.

**Table 1. Activin genes in mouse genome.**

| Gene | Common Name | Ref Seq[1] | P-Accession[2] | Gene ID[3] | Exons[4] |
|---|---|---|---|---|---|
| INHβA | Act A | XM_011244285 | NP_032406 | 16323 | 4 |
| INHβB | Act B | NM_008381 | NP_032407 | 16324 | 2 |
| INHβC | Act C | NM_010565 | NP_034695 | 16325 | 2 |
| INHβE | Act E | NM_008382 | NP_032408 | 16326 | 2 |

[1] NCBI Nucleotide Accession number.

[2] NCBI Protein Accession number. For corresponding human protein data, see S1 Table.

[3] NCBI Gene ID.

[4] Exons were determined using the Blat function, UCSC Genome Browser.

Our results reveal significant differences in expression of Act A protein variants in human compared to mouse. In addition, we have identified a novel HBD in mouse and human Act B that may play an important role in regulating distribution and function of Act B dimers and Act A/Act B heterodimers within the extracellular milieu.

# Results

## Gene organization and RNA expression of Act A

Current genomic databanks indicate that the human and mouse genomes contain 4 activin genes designated as *INHβA*, *INHβB*, *INHβC* and *INHβE*. Of these, the most complex is *INHβA* that spans over 27 kb on chromosome (Chr) 13 in mouse and contains 4 exons (Table 1), while its human counterpart spans 25 kb and contains 7 exons [19]. As in humans [19], mouse *INHβB*, *INHβC* and *INHβE* genes all contain 2 exons (Tables 1 and 2). Interestingly, *INHβC* and *INHβE* reside next to each other on Chr 10 in mouse (Table 2) and Chr 12 in humans [19], suggesting that they arose via gene duplication.

**Table 2. Chromosomal location and size of exons in mouse activin genes.**

| Gene | ID[1] | Exon | Position[2] | Size[3] |
|---|---|---|---|---|
| *INHβA* | 16323 | 1–4 | chr13:16004256–16031609 | |
| | | 1 | chr13:16004256–16008953 | 4698 |
| | | 2 | chr13:16012200–16014532 | 2333 |
| | | 3 | chr13:16017182–16017683 | 502 |
| | | 4 | chr13:16026243–16031609 | 5367 |
| *INHβB* | 16324 | 1–2 | chr1:119415463–119422248 | |
| | | 1 | chr1:119420627–119422248 | 1622 |
| | | 2 | chr1:119415463–119418097 | 2635 |
| *INHβC* | 16325 | 1–2 | chr10:127356322–127370548 | |
| | | 1 | chr10:127370081–127370548 | 468 |
| | | 2 | chr10:127356322–127357829 | 1510 |
| *INHβE* | 16326 | 1–2 | chr10:127349402–127351848 | |
| | | 1 | chr10:127351248–127351848 | 601 |
| | | 2 | chr10:127349402–127351011 | 1610 |

[1] NCBI Gene ID.

[2] Exon position was determined using the Blat function, UCSC Genome Browser.

[3] Size of each Exon in nucleotides.

## Translation of Mouse *INHβA mRNA*

**mActA Exon 3**
RIC*RGEGKKKSKKTCA*GVGRKSRAFKEGNHTTFAAR**M**PLLWLRGFLLASCWIIVRSSPTPGSEGHGSAPDCPS
**CALATLPKDGPNSQPEMVEAVKKHILNMLHLKKRPDVTQPVPKAALLNAIRKLHVGKVGENGYVEIEDDIGRRAE
MNELMEQTSEIITFAES**[G]

**mActA Exon 4**
**TARKTLHFEISKEGSDLSVVERAEVWLFLKVPKANRTRTKVTIRLFQQQKHPQGSLDTGDEAEEMGLKGERSEL
LLSEKVVDARKSTWHIFPVSSSIQRLLDQGKSSLDVRIACEQCQESGASLVLLGKKKKKEVDGDGKKKDGSDG
GLEEEKEQSHRPFLMLQARQSEDHPHRRRRRGLECDGKVNICCKKQFFVSFKDIGWNDWIIAPSGYHANYCEG
ECPSHIAGTSGSSLSFHSTVINHYRMRGHSPFANLKSCCVPTKLRPMSMLYYDDGQNIIKKDIQNMIVEECGCS***S
RQVPEKMDLESPEKTVAK*RKKYKIYELNKTTRKIEIIIIKNPQKKNKNKNQKLN*KQDLMKQMKEDVEKYPKAGLRD
EAVKETGIGGGEGGEERMVYLHFFQNQTDCISFIQTGYCPLSCLLRFPCEPGSLLVYSAVMWVSTTQIIMSRKP*VL
KGQSHPLTQVISMSM*HYLCVFQHTHTHTHTHTHTHTHTHTHTHTRPPTHTHTHTHTHTHTHTHTHTHTHTHTRP
HTQTQRCFHTPHAYTYW*KNNSVQVVTFSFLYHFCHQTKPT*NIENKSGKNERPREEEYQVTFR*GAFDPRTMQP
NRFETVYLKEDKKRDFFVLEVT*LIFIFFKERYLKGICLSICWIQTFLYFVNVVFFIVYY**

**Fig 1. Predicted amino acid sequence of the entire mouse *INHβA* RNA reference sequence (XM_011244285).** For clarity, translation of Act A protein encoded by mouse Exon 3 (mActA Exon 3) and part of mActA Exon 4 (first 1983 out of 5367 total nucleotides) is presented. Exon 3 encodes residues 1–129 (bold type, start of translation, initiation M and red arrow), which continues into Exon 4. *, designates stop codon for the open reading frame, the remaining stop codons are indicated by *; [G] is encoded by a shared codon at the end of Exon 3 and beginning of Exon 4. The translated sequence is identical to accession NP_032406. DNA translation into protein was performed using EMBOSS Transeq (www.ebi.ac.uk/Tools/st/emboss_transeq).

The human *INHβA* gene contains 7 exons that are thought to be alternatively spliced and to elicit 3 distinct Act A protein variants named X1, X2 and X3 [19]. Recently, we verified those findings and demonstrated that the variants are produced by human cells and tissues and that the longer X1 and X2 variants contain a typical HBD in their pro-region, while the shorter X3 variant does not [19]. Since the mouse *INHβA* gene contains 4 exons, it could also potentially give rise to multiple mRNAs and protein variants. To address this question, we translated the entire reference sequence of mouse *INHβA* mRNA Ref Seq (XM_011244285) encompassing the 4 exons and used it as input sequence. This analysis suggested that only one open reading frame of 1275 nucleotides was generated starting in exon 3 and ending in exon 4 (Fig 1). The resulting predicted protein matched the amino acid sequence of the only mouse Act A protein variant in the databanks (NP_032406) (heretofore termed mouse Act A). Interestingly, mouse Act A is nearly identical in both size and sequence (96%) to the shortest human Act A X3 variant, sharing identical signal sequences, splice junctions and furin cleavage sites with human X3 (Fig 2; S1 Table and S1 Fig). Like the human X3 variant, mouse Act A lacks a HBD.

To directly test whether mouse Act A protein is indeed the only gene product expressed in vivo, we isolated total RNA from whole mouse embryos and liver. Full-length cDNAs were prepared and used as template in RT-PCR reactions containing forward primers from each of the 4 mouse *INHβA* exons and an anchored, reverse primer near the end of exon 4 (Fig 3A; see S2 Table for primer sequence). When forward primers derived from exons 1 or 2 were included, no PCR amplicons of expected size (> 1kB) were generated, even after varying reaction conditions, annealing temperature or including DMSO that has been shown to improve heat denaturation of template DNAs and yield of PCR products [20]. In contrast, when forward primers residing in exon 3 were included, PCR products of the correct size were readily obtained (Fig 3B). Sanger sequencing of the excised bands confirmed the identity of these transcripts that contained the predicted ATG start site (position 7146 in Ref Seq: XM_011244285; S2 Fig), encoding a mouse protein virtually identical in sequence to human Act A X3 variant.

**Mouse vs Human Activin A protein alignment**

```
Mouse    1    MPLLWLRGFLLASCWIIVRSSPTPGSEGHGSAPDCPSCALATLPKDGPNSQPEMVEAVKK   60
              MPLLWLRGFLLASCWIIVRSSPTPGSEGH +APDCPSCALA LPKD PNSQPEMVEAVKK
Human    1    MPLLWLRGFLLASCWIIVRSSPTPGSEGHSAAPDCPSCALAALPKDVPNSQPEMVEAVKK   60

Mouse    61   HILNMLHLKKRPDVTQPVPKAALLNAIRKLHVGKVGENGYVEIEDDIGRRAEMNELMEQT   120
              HILNMLHLKKRPDVTQPVPKAALLNAIRKLHVGKVGENGYVEIEDDIGRRAEMNELMEQT
Human    61   HILNMLHLKKRPDVTQPVPKAALLNAIRKLHVGKVGENGYVEIEDDIGRRAEMNELMEQT   120

Mouse    121  SEIITFAESGTARKTLHFEISKEGSDLSVVERAEVWLFLKVPKANRTRTKVTIRLFQQQK   180
              SEIITFAESGTARKTLHFEISKEGSDLSVVERAEVWLFLKVPKANRTRTKVTIRLFQQQK
Human    121  SEIITFAESGTARKTLHFEISKEGSDLSVVERAEVWLFLKVPKANRTRTKVTIRLFQQQK   180

Mouse    181  HPQGSLDTGDEAEEMGLKGERSELLLSEKVVDARKSTWHIFPVSSSIQRLLDQGKSSLDV   240
              HPQGSLDTG+EAEE+GLKGERSELLLSEKVVDARKSTWH+FPVSSSIQRLLDQGKSSLDV
Human    181  HPQGSLDTGEEAEEVGLKGERSELLLSEKVVDARKSTWHVFPVSSSIQRLLDQGKSSLDV   240

Mouse    241  RIACEQCQESGASLVLLGKKKKKEVDGDGKKKDGSDG--GLEEEKEQSHRPFLMLQARQS   298
              RIACEQCQESGASLVLLGKKKKKE +G+GKKK G +G   G +EEKEQSHRPFLMLQARQS
Human    241  RIACEQCQESGASLVLLGKKKKKEEEGEGKKKGGGEGGAGADEEKEQSHRPFLMLQARQS   300

Mouse    299  EDHPHRR**RRR**GLECDGKVNICCKKQFFVSFKDIGWNDWIIAPSGYHANYCEGECPSHIAG   358
              EDHPHRRRRRGLECDGKVNICCKKQFFVSFKDIGWNDWIIAPSGYHANYCEGECPSHIAG
Human    301  EDHPHRR**RRR**GLECDGKVNICCKKQFFVSFKDIGWNDWIIAPSGYHANYCEGECPSHIAG   360

Mouse    359  TSGSSLSFHSTVINHYRMRGHSPFANLKSCCVPTKLRPMSMLYYDDGQNIIKKDIQNMIV   418
              TSGSSLSFHSTVINHYRMRGHSPFANLKSCCVPTKLRPMSMLYYDDGQNIIKKDIQNMIV
Human    361  TSGSSLSFHSTVINHYRMRGHSPFANLKSCCVPTKLRPMSMLYYDDGQNIIKKDIQNMIV   420

Mouse    419  EECGCS   424
              EECGCS
Human    421  EECGCS   426
```

**Fig 2. Protein sequence comparison of mouse Act A (NP_032406; 424 aa) and human Act A X3 variant (NP_002183; 426 aa).** Signal peptide is in bold. Furin cleavage site (RRR) separating the pro-region from mature ligand is highlighted in red, while the mature ligand is in green. Note the high degree of homology between mouse and human proteins, summarized by the amino acid consensus sequence in the middle. Just like human X3 variant, mouse Act A lacks a CW motif/ HBD.

To verify the apparent close similarity of mouse Act A and human Act A X3 variant, we used the I-TASSER server at the University of Michigan [21] to define and illustrate the 3D architecture of the two proteins. As shown in Fig 4, both proteins displayed a characteristic and nearly overlapping structure, with a large N-terminal pro-region and a C-terminal mature ligand in which the wrist helical portion was connected to the two fingers and terminal thumb via the palm [17, 19]. Neither protein contains a CW motif/HBD in the pro-region or mature ligand.

## Analysis of mouse and human activin B

Activins exert their biological functions either as homodimers or heterodimers [1], and it is also well established that many members of the TGFβ protein superfamily interact with HSPGs in the extracellular matrix and cell surface [22]. Thus, the absence of a typical CW motif/HBD in mouse Act A and human X3 variant suggests that intriguingly, these proteins would not utilize HS-dependent restraining mechanisms and could potentially be freer to act. An additional and not exclusionary possibility is raised by the fact that Act A can occur as a heterodimer with Act B. If Act B were to possess a HBD, then heterodimers of Act A and Act B would be able to interact with HS and exploit HS-dependent mechanisms for action. To test these possibilities, we extended our analyses to mouse and human Act B.

In both species, the *INHβB* gene contains 2 exons (Table 1) and displays a single open read-ing frame encoding a protein of 411 amino acids in mouse and 407 amino acids in human (Fig

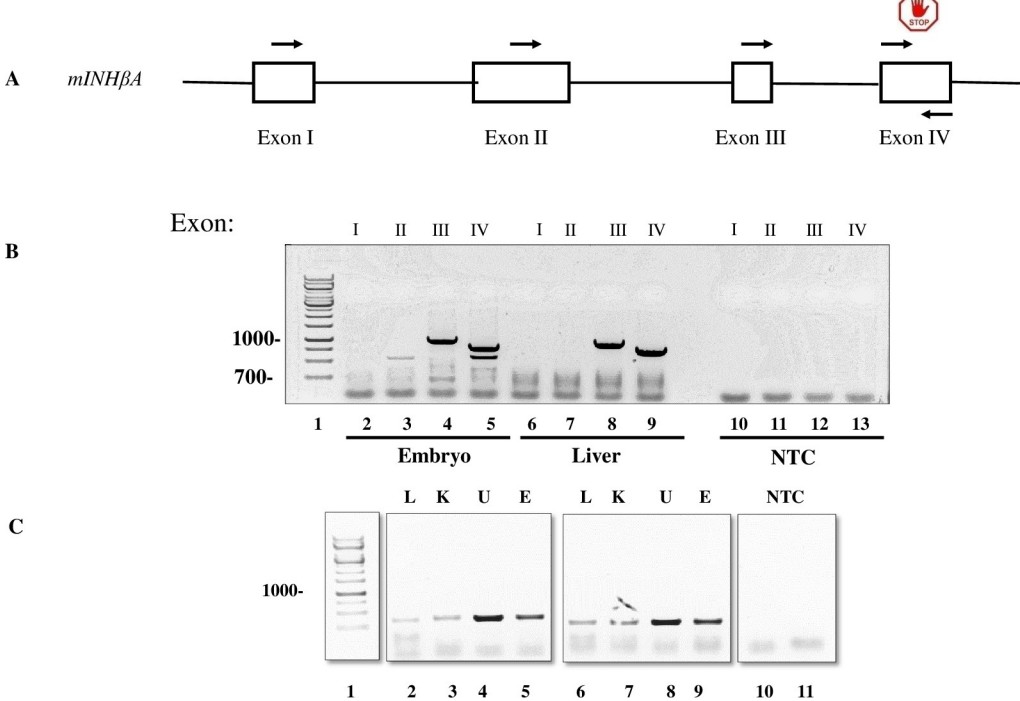

**Fig 3. Organization and expression of mouse *INHβA* gene.** *A*, m*INHβA* gene spans about 27 kB of DNA and contains 4 exons (I-IV) based on current databanks. Forward primers in exons I, II, III and IV are indicated by a top arrow, and an anchored reverse primer in exon IV is indicated by a bottom arrow. *B*, electrophoretic analysis of RT-PCR products. RNAs isolated from whole mouse embryos or liver were processed for cDNA synthesis and subjected to PCR analysis using primers from: exon 1 (primer 1), exon 2 (primer 5), exon 3 (primer 7) or exon 4 (primer 9) along with a common anchored reverse primer (primer 16) annealing near the 3' end of the open reading frame of *INHβA* (all primers are listed in S2 Table). Lane 1, base pair ladder; lanes 2–5, whole embryo RNA PCR products; lanes 6–9, liver RNA PCR products; and lanes 10–13, no template controls (NTC). Note no PCR products were generated with forward primers derived from exons 1 and 2, while amplicons of the correct size were produced when forward primers from exons 3 and 4 used (lanes 4, 5, 8 and 9). *C*, electrophoretic analysis of RT-PCR products derived from mouse *INHβB*. cDNA libraries were generated from embryo (E), liver (L), kidney (K), uterus (U) and used as template for PCR reactions using primers (see S2 Table) from: exon 1 (primer 20 or 21), lanes 2–5 and 6–9, respectively and a common anchored reverse primer (primer 22) annealing near the 3' of exon 2 of *INHβB* mRNA. Note that Act B is highly expressed in embryonic tissue (E) and uterus (U) and at lower levels in liver (L) and kidney (K). The identity of the PCR products was confirmed by Sanger DNA sequencing (S2 Fig). NTC, no template control.

5). The proteins have an identical signal peptide, a consensus furin splicing site (RKR) [8], a large N-terminal pro-region and a mature ligand (Fig 5 and S1 Table). Attentive survey of their amino acid sequences quickly revealed the presence of the short amino acid segment with the sequence KGSRRKVRVK in the pro-region of each protein (residues 195–204 in mouse and 191–200 in human). This segment is rich in basic residues and closely resembles a typical HS-binding CW motif with a consensus sequence BXXBBBXBXB, where B is a basic residue (Lys or Arg) and X is an uncharged residue [12, 15].

To determine whether this motif could serve as part of a HBD within Act B, we synthesized a peptide encompassing the motif and included a biotin tag on the N-terminus via a Gly linker to facilitate analysis (Fig 6A). The peptide was next tested in solid-phase binding assays where HS was immobilized on standard 96-well plates, following procedures we have reported previously [23]. The peptide did interact with HS with high affinity and saturable binding kinetics ($K_d \sim 8$ nM), and binding was fully prevented by addition of soluble heparin competitor (Fig 6C). Stereological analysis indicated that the motif exhibited an unstructured configuration (Fig 6B).

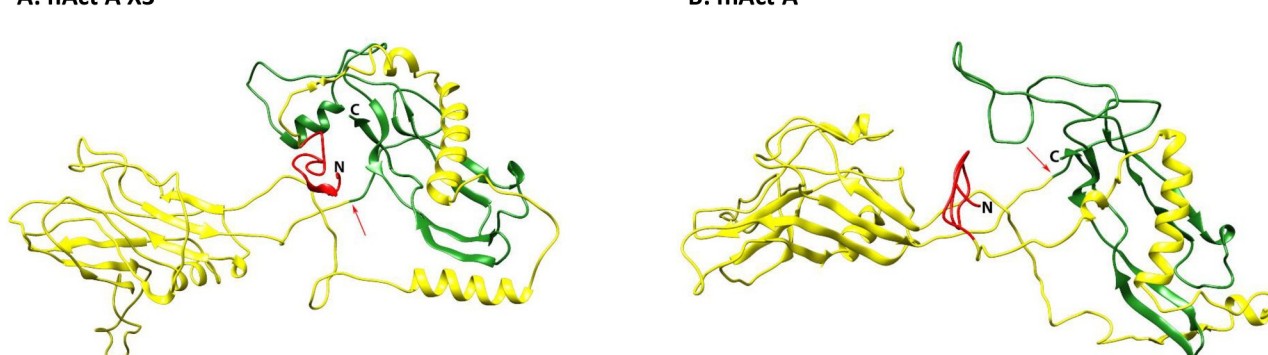

**Fig 4. Modeling of mouse Act A and human Act A X3 variant.** Protein structure predictions were generated using the I-TASSER server for Protein Structure and Function. (Left) human Act A X3 variant (accession # NP_002183). (Right) mouse Act A (accession # NP_032406). Note the nearly identical 3D configuration of the two proteins. The large and highly structured N-terminal pro-region (yellow) continues with the C-terminal mature ligand (green). The signal peptide on the N-terminus is in red while the Furin cleavage site is at the arrow.

The ability of this peptide to bind immobilized HS is an accepted test of interaction specificity, but it does not reproduce the more complex process of cell or tissue interactions. Thus, we determined whether the peptide was able to interact with the cell surface. As in previous studies, we prepared HBCs consisting of the peptide complexed with a fluorescent NA derivative (NA-488) [19]. The resulting complex was incubated with U937 cells that had been lightly pre-fixed with 1% buffered formalin. After incubation, the cells were washed, and the amount of bound peptide was assessed by FACs. Clearly, the peptide was able to readily interact with the cell surface, and its binding was blocked by soluble heparin competitor (Fig 6D). Thus, mouse

## Mouse vs Human Activin B protein alignment

```
Mouse    1   MDGLPGRALGAACLLLLVAGWLGPEAWGSPTPPPSPAAPPPPPPPGAPGGSQDTCTSCGG   60
             MDGLPGRALGAACLLLL AGWLGPEAWGSPTPPP+PAAPPPPPPPG+PGGSQDTCTSCGG
Human    1   MDGLPGRALGAACLLLLAAGWLGPEAWGSPTPPPTPAAPPPPPPPGSPGGSQDTCTSCGG   60

Mouse    61   GGGGFRRPEELGRVDGDFLEAVKRHILSRLQLRGRPNITHAVPKAAMVTALRKLHAGKVR   120
                  FRRPEELGRVDGDFLEAVKRHILSRLQ+RGRPNITHAVPKAAMVTALRKLHAGKVR
Human    61   ----FRRPEELGRVDGDFLEAVKRHILSRLQMRGRPNITHAVPKAAMVTALRKLHAGKVR   116

Mouse    121   EDGRVEIPHLDGHASPGADGQERVSEIISFAETDGLASSRVRLYFFVSNEGNQNLFVVQA   180
               EDGRVEIPHLDGHASPGADGQERVSEIISFAETDGLASSRVRLYFF+SNEGNQNLFVVQA
Human    117   EDGRVEIPHLDGHASPGADGQERVSEIISFAETDGLASSRVRLYFFISNEGNQNLFVVQA   176

Mouse    181   SLWLYLKLLPYVLEKGSRRKVRVKVYFQEQGHGDRWNVVEKKVDLKRSGWHTFPITEAIQ   240
               SLWLYLKLLPYVLEKGSRRKVRVKVYFQEQGHGDRWN+VEK+VDLKRSGWHTFP+TEAIQ
Human    177   SLWLYLKLLPYVLEKGSRRKVRVKVYFQEQGHGDRWNMVEKRVDLKRSGWHTFPLTEAIQ   236

Mouse    241   ALFERGERRLNLDVQCDSCQELAVVPVFVDPGEESHRPFVVVQARLGDSRHRIRKRGLEC   300
               ALFERGERRLNLDVQCDSCQELAVVPVFVDPGEESHRPFVVVQARLGDSRHRIRKRGLEC
Human    237   ALFERGERRLNLDVQCDSCQELAVVPVFVDPGEESHRPFVVVQARLGDSRHRIRKRGLEC   296

Mouse    301   DGRTSLCCRQQFFIDFRLIGWNDWIIAPTGYYGNYCEGSCPAYLAGVPGSASSFHTAVVN   360
               DGRT+LCCRQQFFIDFRLIGWNDWIIAPTGYYGNYCEGSCPAYLAGVPGSASSFHTAVVN
Human    297   DGRTNLCCRQQFFIDFRLIGWNDWIIAPTGYYGNYCEGSCPAYLAGVPGSASSFHTAVVN   356

Mouse    361   QYRMRGLNPGPVNSCCIPTKLSSMSMLYFDDEYNIVKRDVPNMIVEECGCA   411
               QYRMRGLNPG VNSCCIPTKLS+MSMLYFDDEYNIVKRDVPNMIVEECGCA
Human    357   QYRMRGLNPGTVNSCCIPTKLSTMSMLYFDDEYNIVKRDVPNMIVEECGCA   407
```

**Fig 5. Comparison of mouse Act B (NP_032407; 411 aa) and human Act B (NP_002184; 407 aa) protein sequences.** Signal peptide is denoted in bold. Furin cleavage site (RKR) at end of pro-region is in red and the mature ligand is in green. The CW motif part of the putative HBD is in blue. Note the high degree of homology between mouse and human proteins.

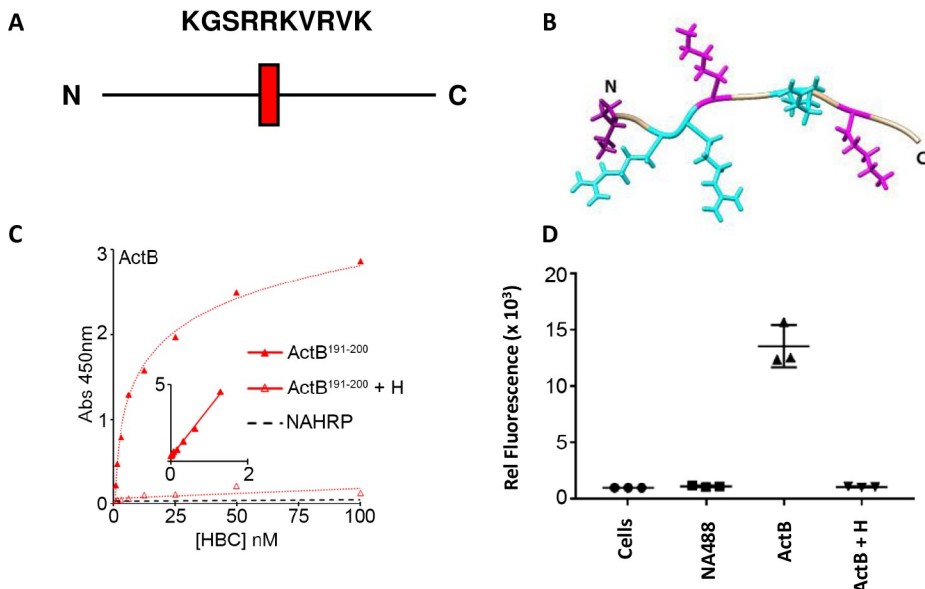

**Fig 6. Peptide interactions with HS and cell surface.** *A*, Sequence of the HBD in Act B. Note the CW motif is centrally located in the pro-region of mouse (residues 195–204) and human (residues 191–200) Act B. *B*, Stereological structure of the motif was generated using the I-TASSER server for Protein Structure and Function. Note that the peptide has a disordered conformation, with no distinct secondary structure. *C*, solid-phase binding assays. A biotinylated peptide encompassing the CW motif was mixed with NA-HRP to form a heparin-binding complex (HBC) and subsequently incubated with immobilized HS. After incubation, plates were washed and developed with chromogenic substrate. Note that the HBC bound to HS with saturable kinetics. Inset, double reciprocal plots showing high affinity binding ($K_d$ ~ 8 nM). *D*, FACs analysis. U937 cells were incubated with HBCs for 2 hr and then washed, and bound peptide was assessed by FACs. Statistical analysis (Student's T test): *, NA488 vs Act B: $p < 0.004$; **, Act B vs Act B + H: $p < 0.002$. Note that the peptide binds to the cells and binding is blocked by addition of soluble heparin competitor.

and human Act B contain a typical HBD that exhibits high affinity binding to HS and the cell surface.

Next, we tested the ability of full-length Act B protein (pro-region and mature ligand) to interact with HS. An expression construct containing the entire open reading frame of Act B with a His tag on the C-terminus was transfected into AD-293 cells (Fig 7) as described [19]. After 3 days, conditioned medium from the transfected cells was harvested and incubated with SP Sepharose (a strong cation exchange resin), Talon resin (interacts with the His tag) or heparin agarose. After incubation, the resins were washed, and bound proteins were eluted, size fractionated on SDS-polyacrylamide gels and analyzed by immunoblot. We found that a protein of ~ 14 kDa (the predicted size of the mature Act B ligand) bound to SP Sepharose and Talon resins and was recognized by anti-Act B or anti-His antibodies (Fig 7C and 7D). We also observed some higher molecular weight bands on these blots (brackets, Fig 7C and 7D) that likely represented unprocessed forms of the pro-protein and were absent in conditioned medium from control non-transfected cells. Interestingly, in addition to binding to Talon, these higher mass proteins were able to interact with heparin agarose (Fig 7D, lane 2). This is consistent with the fact that the HBD resides in the pro-region (see above) of Act B. Indeed, the mature Act B ligand (Fig 7D, arrow on right) did not interact with heparin agarose (Fig 7D, compare lanes 2 and 3).

To obtain additional insights into how Act B interacts with HS, we used the I-TASSER server [24] to interrogate the overall structure of full-length Act B and assess the location and orientation of the HBD with respect to the entire protein (Fig 8). We found that mouse and

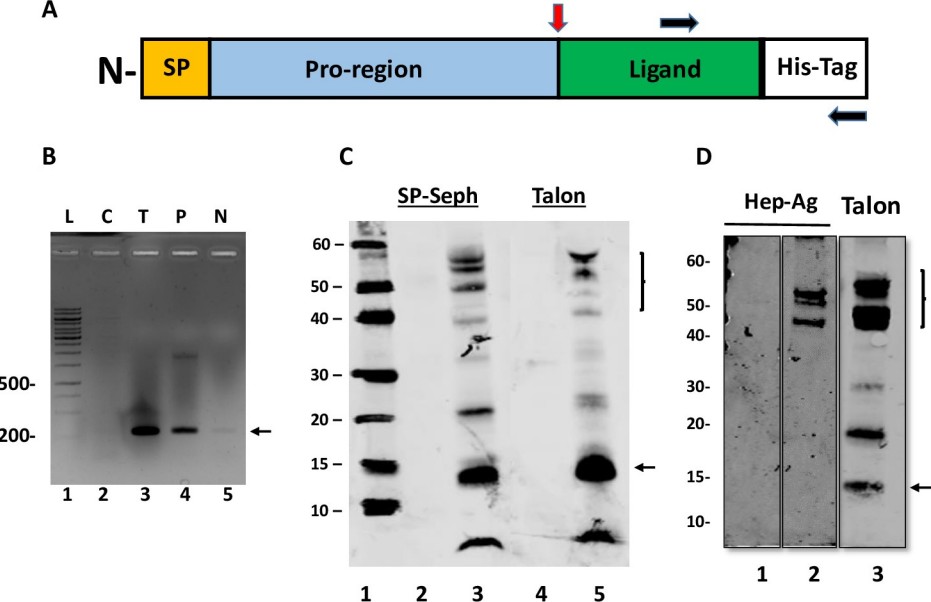

**Fig 7. Act B protein expression and interaction analyses.** *A*, schematic of the Act B construct cloned into expression vector pcDNA 3.1. Red arrow denotes furin cleavage site; black arrows designate primers used for RT-PCR, yielding an amplicon of 267 bp (arrow on right in *B*). *B*, electrophoretic analysis of RT-PCR products elicited by Act B expression construct. cDNAs prepared from control and transfected AD-293 cells and used as template for RT-PCR. Lane 1, bp ladder (L); lane 2, control (untransfected) cells (C); lane 3, cells transfected with Act B expression construct (T); lane 4, positive control, Act B-pcDNA 3.1 expression construct used as template (P); lane 5, no template control (N). *C* and *D*, Act B protein expression analyses. Conditioned medium samples were collected from AD-293 cells that were non-transfected or transfected with Act B expression vector and incubated with SP-Sepharose (lanes 2, 3 in C) or Talon resin (lanes 4, 5 in C and lane 3 in D) or heparin-agarose (lanes 1, 2 in *D*), and bound proteins were analyzed on immunoblots probed with anti-Act B antibodies. Note the presence of both mature Act B ligand (with predicted mass of ~14 kDa, arrow) and several higher molecular weight bands (brackets) that bound SP sepharose and Talon (lanes 3 and 5 in C and lane 3 in D.). In *C*: lane 1, Novex Sharp Pre-stained markers; lanes 2 and 4, conditioned medium from control cells; lanes 3 and 5, conditioned medium from cells transfected with Act B expression construct. In *D*: conditioned medium from control cells (lane 1) or cells transfected with Act B expression construct incubated with Heparin-agarose (lane 2) or Talon resin (lane 3). Note that high molecular weight proteins bind both resins, while mature Act B ligand (14 kDa; arrow on right) does not bind heparin (lane 2) but binds Talon (lane 3). Numbers on left in D, Mass markers in kDa.

human Act B proteins exhibited a very similar three-dimensional structure (Fig 8), in close correlation with their 96% primary sequence homology (Fig 5). Interestingly, the CW motif/ HBD appeared to be located on the surface of the pro-region, presumably well exposed for interactions with HS, but displayed no obvious ordered structure or distinct secondary structure.

## Discussion

Our genomic, RNA and protein expression analyses provide new and extensive insights into both similarities and differences in activin genes and protein variant expression in mice and humans. Both species contain 4 activin genes, *INHβA*, *INHβB*, *INHβC* and *INHβE* of which *INHβA* is the most complex and contains 4 exons in mouse and 7 exons in humans, while the other 3 genes contain only two exons. In an earlier report, we demonstrated by RNA analyses on human cells and tissues that the *INHβA* gene is alternatively spliced and generates three Act A protein variants (X1, X2 and X3: longest to shortest) [19]. These variants contain a divergent N-terminal pro-region, but the same C-terminal mature ligand. We found that these proteins were not only synthesized, but also processed and secreted properly since we were

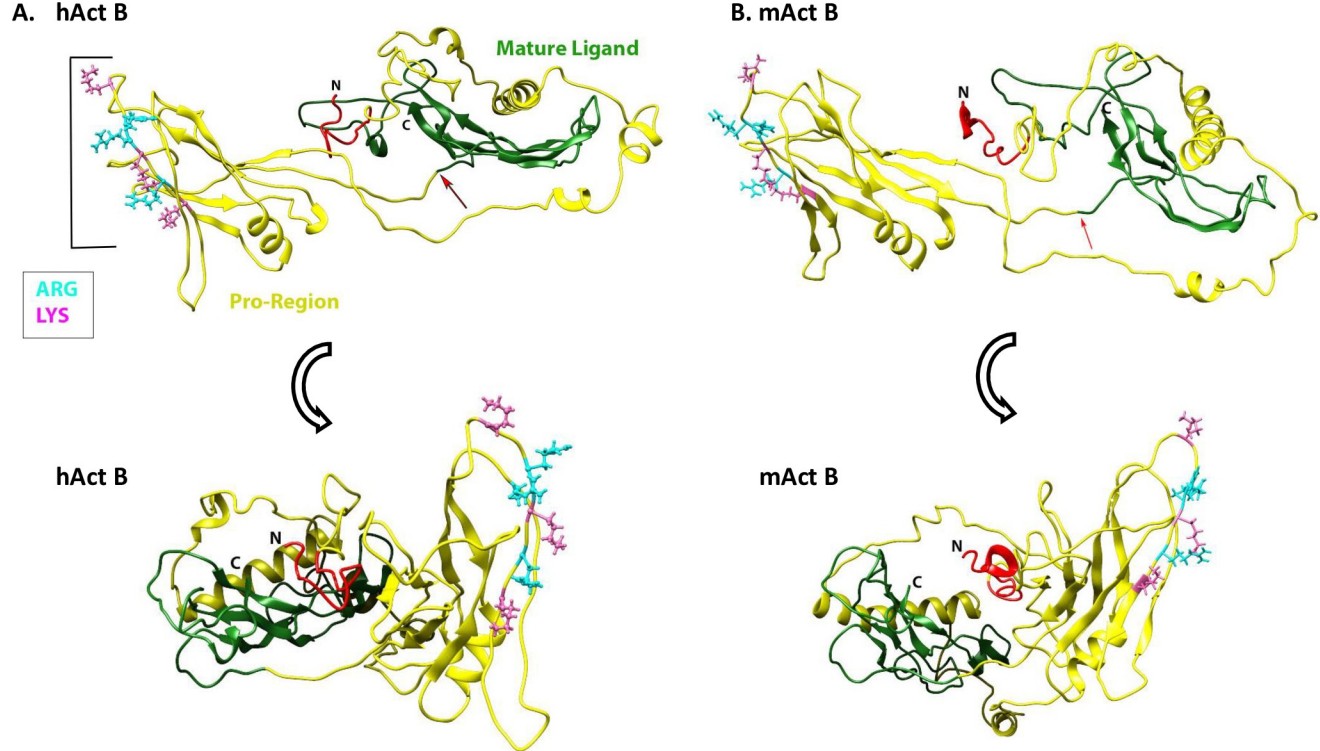

**Fig 8. Modeling of full-length Act B.** Protein structure predictions were generated using the I-TASSER server. Note that human and mouse Act B proteins (hAct B and mAct B, respectively) display similar architecture with a large pro-domain (in yellow) and mature ligand (in green). The CW motif/HBD, rich in Arg and Lys resides, appears to reside on the lateral surface, approximately in the middle of the pro-region, and is delineated by a bracket in hAct B. NCBI accession numbers: hAct B, NP_002184; mAct B, NP_032407.

able to recover mature human Act A ligand (116 residues) following transfection of expression constructs containing full-length cDNAs into AD-293 cells [19]. In contrast, we find here that only one Act A transcript can be amplified using mouse cDNA as template in RT-PCR reactions. This RNA begins at a predicted AUG codon in exon 3 and ends in exon 4 and encodes a 424 amino acid protein that is essentially identical (96%) to the shortest human X3 variant (see Figs 1–3). Clearly the difference in *INHβA* gene structure and exon numbers in mouse and human genomes elicits significant differences in Act A mRNA and protein expression. It is interesting to note that the rat *INHβA* gene contains 5 exons (transcript variants X4, XM_008771714; X5, XM_008771715 and X6, XM_008771716), thus seemingly representing an intermediate evolutionary step between mouse with 4 exons and humans with 7 exons. It is possible that such changes in gene structure and expression reflect evolutionary modulations in the range and function by Act A and its variants in these species.

It is generally assumed that activins are synthesized, dimerize and undergo proteolytic cleavage by furin/proprotein convertases intracellularly and are subsequently secreted as complexes containing the N-terminal pro-region non-covalently associated with the C-terminal mature ligand. Following secretion, this complex binds HSPGs or other ECM components and the mature ligand is released by an unknown mechanism [17]. In the current report, we show that mouse Act A protein, like its human X3 counterpart lack a HBD. In earlier studies, we found that the longer human Act A X1 and X2 variants do possess a stereotypic HBD near the N-terminus of their pro-region with high affinity binding capacity for HS in solid-phase and cell surface binding assays [19]. One would predict that the absence or presence of a HBD

would have functional consequences since the expected interactions of human Act A X1 and X2 isoforms with HSPGs would influence protein distribution and activity. Most previous studies on human Act A have made use of full-length X3 variant expression constructs [25] and thus, some of the data and insights may have to be re-evaluated using X1 or X2 constructs. Because the only mouse Act A also lacks a HBD as we show here, how would it exist within the extracellular environment and would it have other means of interacting with the ECM? As suggested above, one possible answer is offered by the fact that Act A exists either as a homodimer or a heterodimer with Act B. Thus, it is possible that mouse Act A dimers (and human X3 dimers) do not interact with HSPGs and are free to exert their biological activity. In contrast, an ActA-ActB heterodimer could exploit the HBD in Act B reported here, to interact with HSPGs and exhibit more restrained and perhaps more selective biological action and range in vivo.

In their seminal studies, Cardin and Weintraub first described the domains present in four proteins- apolipoprotein B, apolipoprotein E, vitronectin and platelet factor-4- that mediate interactions with HS. In these studies, two binding motifs were identified–XBBXBX and XBBBXXBX–where B represents a basic residue (Arg or Lys) and X represents a non-charged amino acid [15]. Additional motifs have since been described in many proteins [12] and are frequently referred to as Cardin Weintraub (CW) motifs. The CW motif we identified in Act B- KGSRRKVRVK- has a consensus BXXBBBXBXB, and binds HS with high affinity in solid-phase binding assays and binds to the cell surface (Fig 6). In silico modeling reveals that the motif has a disordered structure resembling the HBD present on the C-terminal region of mature BMP 5–7, while differing from the HBD present in the N-terminal region of mature BMP 2–4, which has a helical configuration [23]. Our modeling of full-length Act B protein shows that its HBD resides on the surface, in the middle of the pro-region, where it would be readily available to interact with HSPGs and potentially other components in the ECM. Similarly, the HBD present in human Act A X1 and X2 variants also resides in the pro-region, but differs from the domain in Act B (Fig 8) as it has a helical structure and resides closer to the N-terminus of the pro-region [19]. Clearly, different members of the TGFβ superfamily have evolved HBDs which differ in secondary structure and location either within the pro-region or mature ligand, while some lack a HBD such as mouse Act A and the human Act A X3 variant. Future studies will need to be directed at uncovering the seemingly important implications of this broad diversity. It should also be remembered that while HBDs are important mediators of protein-HS interactions and protein distribution and function, there is evidence that other individual residues or short segments have roles in these interactions, as well [26].

Activins are multi-functional cytokines which play important roles in a wide range of normal and abnormal physiological processes. One example is Fibrodysplasia Ossificans Progressiva (FOP), a severe congenital disorder characterized by formation and accumulation of extra-skeletal bone (heterotopic ossification) [27]. The majority of FOP patients carry mutations in the type I BMP receptor ALK2 encoded by *ACVR1* and display a recurrent *ACVR1*[R206H] mutation [27]. Under normal circumstances, ALK2 interacts with various BMPs and signals via canonical SMAD1/5/8 pathway, but is not responsive to activins [6, 28]. Recent important studies have revealed that mutant ALK2[R206H] becomes responsive to exogenous Act A and elicits pSMAD1/5/8 signaling [29]. Treatment of FOP-like mutant mice with an Act A neutralizing monoclonal antibody was found to significantly inhibit heterotopic ossification [29]. Our data here show that mouse Act A does not possess a HBD and thus, may have a broader activity range. In contrast, an Act A-Act B heterodimer could exploit the HBD present in the pro-region of Act B and interact with HSPGs. Lastly, it remains to be studied whether FOP pathogenesis is influenced by expression and/or bioavailability of Act A homodimers

**Table 3. Primers used to amplify INHβB transcripts in AD-293 cells with expression constructs.**

| Primer | Target | Accession[1] | Sequence (5'-3') | Region[2] |
|---|---|---|---|---|
| 1 | hINHβB | NM_002193 | F: GCTACTACGGGAACTACTGT | 1030–1049 |
| 2 | hINHβB | NM_002193 | F: TGTACTTCGATGATGAGTACAACAT | 1201–1225 |
| 3 | hINHβB | | R: CTAATGATGGTGGTGATGATGG | His Tag (3'end) |

[1] NCBI accession number.

[2] Annealing position of primer.

versus heterodimers and whether the significant differences in Act A variants in humans and mice could affect disease progression, severity and responses to treatment.

## Experimental procedures

### Reagents

NeutrAvidin (NA), NA-HRP, NA-DyLight 488 (NA-488) and Fisher exACTGene DNA Ladder (100–10,000 bp) were obtained from Thermo-Fisher. Human Act B Antibody (MAB659) was obtained from RD Systems. Heparin was obtained from Sagent Pharmaceuticals and HS was obtained from Millipore-Sigma (www.sigmaaldrich.com). All DNA Oligonucleotide primers were obtained from Integrated DNA Technologies (www.idtdna.com) and are listed in Table 3 and S2 Table. Peptides were synthesized and purified by Peptide 2 (www.peptide2.com).

### Tissue isolation from mice

All studies were approved by the Institutional Care and Use Committee (IACUC), Children's Hospital of Philadelphia under Protocol # IAC 17–000958.

### RNA, cDNA and DNA sequencing

RNA was isolated from adult mouse tissues and embryos using TRIzol Reagent (Thermo-Fisher) following the manufacturers protocol. Five micrograms of glycogen was added to the aqueous phase prior to addition of isopropanol to facilitate RNA precipitation. RNA concentration was determined using a Nanodrop spectrophotometer. cDNA was prepared with a Versco cDNA synthesis kit, using 3 μg of RNA as template plus 200 ng of random hexamer and 200 ng oligo dT as primers, following the manufacturer's instructions. All PCR reactions were optimized for each primer set by gradient PCR, using an Applied Biosystems Verity thermal cycler. PCR reactions contained 1 μl of cDNA, primer sets (S2 Table) and GoTaq Green master mix (Promega) following the manufactures protocol. PCR products were size fractionated on 1% agarose gels and subsequently the bands were isolated, DNA was purified from the gel using Micro Bio-Spin columns (Bio-Rad) and subjected to sequencing. All Sanger DNA sequencing was performed by the Center for Applied Genomics (CAG; caglab.org) at the Children's Hospital of Philadelphia.

### Solid-phase binding assays

Nunc MaxiSorp 96 well flat bottom plates were coated with HS in 50 mM carbonate buffer (pH 9.4) overnight at 4°C as described [23]. All binding assays were carried out in PBS, 0.1% tween 20 (PBST) containing 1% bovine serum albumin. The plates were incubated with peptide tetramers consisting of biotinylated peptide complexed with NA-HRP for a minimum of 2

hr at room temperature with gentle shaking. At the termination of the assay, the plates were washed 3 times with PBST and a final rinse with PBS. Plates were developed by addition of HRP substrate O-phenylenediamine dihydrochloride (OPD) in Citrate-Phosphate buffer (25 mM citric acid, 50 mM sodium phosphate, pH 5) and read at 450 nm in a Bio-Tek Synergy HT plate reader. Very low levels of background binding were exhibited by NA-HRP alone or when complexed with biotinylated-BSA.

## Peptide binding to intact cells

U937 (human monocytic-like) cells were washed with PBS and fixed with 1% buffered formalin for 20 min on ice. The cells were washed with PBS and blocked by incubation in PBS, 1% BSA for 20 min on ice. Approximately $10^6$ cells (100 μl) were incubated with peptide tetramers consisting of biotinylated peptide complexed with NA-488, to form a heparin binding complex. Following incubation for 2 hr on ice, the cells were washed and analyzed on a BD Accuri Flow Cytometer (FL1 channel), located in the Flow Cytometry Core Laboratory at CHOP [19]. Statistical evaluation of data was performed using Student's t-test employing GraphPad Prism software (www.graphpad.com).

## Activin B protein expression constructs

The entire open reading frame for human Act B was cloned into EcoR1/Xba1 sites of mammalian expression vector pcDNA3.1 as described [19]. This expression construct contains a cytomegalovirus (CMV) promoter for constitutive expression in mammalian cells. A His$_8$ tag was added to the C-terminus (3'end of cDNA) to facilitate purification via immobilized metal affinity chromatography (IMAC). After cloning, the resulting expression constructs were verified by Sanger DNA sequencing. For protein expression, 6 μg of plasmid DNA plus 18 μl of FuGene6 Transfection reagent (Promega) was added to 600 μl of Opti-MEM medium and incubated for 20 minutes at room temperature. Next, the transfection mixture was added to AD-293 cells growing in 10 ml RPMI medium and 5% fetal calf serum. Conditioned medium was collected after 72 hours and incubated with TALON metal affinity resin (Takara Bio USA) for purification. The resin was washed with PBS and transferred to Micro Bio-Spin columns (Bio-Rad). His-Tag bound proteins were eluted with 0.5 M imidazole and analyzed on NuPAGE 4–12% Bis-Tris Gels (Invitrogen) under reducing conditions followed by transfer to PVDF membranes. Blots were probed with primary anti-His and anti-Act B antibodies and developed with IRDye Secondary Antibodies (Li-Cor).

## mRNA expression of Act B constructs

To evaluate mRNA expression of Act A expression constructs, transfected cells were washed and pelleted. RNA was isolated by combining Trizol and Direct-zol RNA MiniPrep Kit (Zymo Research #2050) following the manufacturers protocols. During isolation, RNA is treated with DNAase, to remove any contaminating plasmid or genomic DNA. RNA concentration and purity was assessed using a NanoDrop 2000 spectrophotometer. Act B mRNA expression was determined using forward primer 1 and reverse primer 3 (Table 3) by RT-PCR, which yields an amplicon of 267 bp.

## Sequence alignments and in silico modeling

Alignments of primary sequences were constructed using Unipro UGENE [30]. Protein structure predictions were generated using the I-TASSER server for Protein Structure and Function

at the University of Michigan (zhanglab.ccmb.med.umich.edu) [21]. The resulting structures were visualized using Chimera modeling software (www.cgl.ucsf.edu/chimera).

## Statistical analysis

Statistical evaluation of data utilized Student's t-test employing GraphPad Prism software (www.graphpad.com).

## Supporting information

**S1 Fig. Alignment of Mouse and Human mature Activin ligands.**
(PPTX)

**S2 Fig. Sanger Sequencing of purified PCR products from Fig 3.**
(PPTX)

**S3 Fig. Mouse vs Human Activin C protein alignment.**
(PPTX)

**S4 Fig. Mouse vs Human Activin E protein alignment.**
(PPTX)

**S5 Fig. Uncropped western blots shown in Fig 7 of the manuscript.**
(PPTX)

**S1 Table. General features of mouse (m) and human (h) Activin proteins.**
(DOCX)

**S2 Table. Primers used to amplify Murine INHβA and INHβB transcripts.**
(DOCX)

## Acknowledgments

We thank the Flow Cytometry Core Laboratory at CHOP for helpful assistance.

## Author Contributions

**Conceptualization:** Paul C. Billings, Maurizio Pacifici.

**Data curation:** Paul C. Billings, Candice Bizzaro, Evan Yang, Juliet Chung, Christina Mundy.

**Formal analysis:** Paul C. Billings, Evan Yang, Christina Mundy.

**Funding acquisition:** Maurizio Pacifici.

**Methodology:** Paul C. Billings, Christina Mundy.

**Supervision:** Paul C. Billings, Christina Mundy, Maurizio Pacifici.

**Writing – original draft:** Paul C. Billings.

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
