## [Decision Letter · Decision Letter 0]

30 Jan 2020

PONE-D-20-00545

Human and mouse activin genes: divergent expression of activin A protein variants

and identification of a novel heparan sulfate-binding domain in activin B

PLOS ONE

Dear Dr. Billings,

Thank you for submitting your manuscript to PLOS ONE. After careful consideration, we feel that it has merit but does not fully meet PLOS ONE’s publication criteria as it currently stands. Therefore, we invite you to submit a revised version of the manuscript that addresses the points raised during the review process.

We would appreciate receiving your revised manuscript by Mar 15 2020 11:59PM. To enhance the reproducibility of your results, we recommend that if applicable you deposit your laboratory protocols in protocols.io, where a protocol can be assigned its own identifier (DOI) such that it can be cited independently in the future. For instructions see: http://journals.plos.org/plosone/s/submission-guidelines#loc-laboratory-protocols

We look forward to receiving your revised manuscript.

Kind regards,

Xin Zhang, Ph.D.

Academic Editor

PLOS ONE

Journal Requirements:

Reviewers' comments:

Reviewer's Responses to Questions

**Comments to the Author**

1. Is the manuscript technically sound, and do the data support the conclusions?

Reviewer #1: Partly

Reviewer #2: Yes

2. Has the statistical analysis been performed appropriately and rigorously? 

Reviewer #1: N/A

Reviewer #2: Yes

3. Have the authors made all data underlying the findings in their manuscript fully available?

Reviewer #1: Yes

Reviewer #2: Yes

4. Is the manuscript presented in an intelligible fashion and written in standard English?

Reviewer #1: No

Reviewer #2: Yes

5. Review Comments to the Author

Reviewer #1: This MS entitled: Human and mouse activin genes: divergent expression of activin A protein variants and identification of a novel heparan sulfate-binding domain in activin B” by Billings et al reports the gene expression of Activins in mouse. It is a related work to the previous published paper (Identification and characterization of a novel heparan sulfate-binding domain in Activin A longest variants and implications for function published in the same journal 2019). The data are clear and the paper is well-written.

Reviewer #2: In this study, the authors investigated the differential expression of Activin protein variants in humans and mice, and analyzed their interaction with HS. They reported the absence of HBD in mouse Act A, which is instead identified in mouse and human Act B. These findings imply a novel and potentially important mechanism that regulates the distribution and function of activin (hetero-)dimers within the extracellular milieu. Overall the manuscript is well written and the data are appropriately presented. I have only the following points that the authors can consider commenting on or discussing in more detail.

Major point

1. In Act B, HBD is located in the pro-region, not the mature ligand. Considering the localization of HS chains on cell surface, it is important for the authors to clarify whether the dimers are secreted as full-length protein with mature ligand still associated with the pro-region after intracellular cleavage by furin/proprotein convertases. Or, whether the pro-regions interact with the ECM alone or with the mature growth factor after secretion.

Minor points

1. [Abstract] “Activins are synthesized and secreted as large complexes made of a long pro-region and a short mature N-terminal ligand” and [Introduction] “resulting in a long C-terminal pro-region and a N-terminal active mature ligand”: should be N-terminal pro-region and a C-terminal mature ligand.

2. Fig. 4: Annotation for the colors of the protein structures is desired.

3. Typo: P16, L22: “sold” should be “solid”.

6. PLOS authors have the option to publish the peer review history of their article (what does this mean?). If published, this will include your full peer review and any attached files.

Reviewer #1: No

Reviewer #2: No

---

## [Author Response · Author response to Decision Letter 0]

31 Jan 2020

January 31, 2020

RE: PONE-D-20-00545- Human and mouse activin genes: divergent expression of activin A protein variants and identification of a novel heparan sulfate-binding domain in activin B

Xin Zhang, Ph.D.

Academic Editor

PLOS ONE

Dear Dr. Zhang:

Thank you for your prompt review of our manuscript. We have enclosed our revised manuscript and have addressed the comments of each Reviewer and Editor below. Corrections and changes in the revised manuscript are in bold type. 

Reviewer #1: This MS entitled ‘Human and mouse activin genes: divergent expression of activin A protein variants and identification of a novel heparan sulfate-binding domain in activin B’ by Billings et al reports the gene expression of Activins in mouse. It is a related work to the previous published paper (Identification and characterization of a novel heparan sulfate-binding domain in Activin A longest variants and implications for function published in the same journal 2019). The data are clear and the paper is well-written.

Response

We thank Reviewer #1 for their constructive evaluation of our work. 

Reviewer #2: In this study, the authors investigated the differential expression of Activin protein variants in humans and mice, and analyzed their interaction with HS. They reported the absence of HBD in mouse Act A, which is instead identified in mouse and human Act B. These findings imply a novel and potentially important mechanism that regulates the distribution and function of activin (hetero-) dimers within the extracellular milieu. Overall the manuscript is well written and the data are appropriately presented. I have only the following points that the authors can consider commenting on or discussing in more detail.

Major point

1. In Act B, HBD is located in the pro-region, not the mature ligand. Considering the localization of HS chains on cell surface, it is important for the authors to clarify whether the dimers are secreted as full-length protein with mature ligand still associated with the pro-region after intracellular cleavage by furin/proprotein convertases. Or, whether the pro-regions interact with the ECM alone or with the mature growth factor after secretion.

Minor points

1. [Abstract] “Activins are synthesized and secreted as large complexes made of a long pro-region and a short mature N-terminal ligand” and [Introduction] “resulting in a long C-terminal pro-region and a N-terminal active mature ligand”: should be N-terminal pro-region and a C-terminal mature ligand.

2. Fig. 4: Annotation for the colors of the protein structures is desired.

3. Typo: P16, L22: “sold” should be “solid”. 

Response

We thank Reviewer #2 for their careful and constructive evaluation of our work. 

Major point

1. As requested by the Reviewer, we have addressed the issue of Activin secretion and have added the following statement to the Discussion section. 

It is generally assumed that activins are synthesized, dimerize and undergo proteolytic cleavage by furin/proprotein convertases intracellularly and are subsequently secreted as complexes containing the N-terminal pro-region non-covalently associated with the C-terminal mature ligand. Following secretion, this complex binds HSPGs or other ECM components and the mature ligand is released by an unknown mechanism [17]. 

Minor points

1. As suggested by the reviewer, we have revised the statement in the Abstract to read “Activins are synthesized and secreted as large complexes made of a long pro-region and a short mature C-terminal ligand”.

We have corrected the statement in the Introduction to read “resulting in a long N-terminal pro-region and a C-terminal mature ligand.

2. Fig 4, legend. As requested, we have indicated the colors of the protein structures as follows:

The large and highly structured N-terminal pro-region (yellow) continues with the C-terminal mature ligand (green). The signal peptide on the N-terminus is in red while the Furin cleavage site is at the arrow.

3. Typo: P16, L22: As requested, “sold” has been corrected to “solid”. 

Editorial Comments

1. As requested by the Editor, we have included uncropped images of gels in Supporting information- S5 Fig. Uncropped western blots shown in Fig 7 of the manuscript.

2. As requested, the section on use of animal tissue has been revised and the following statement has been added to the Methods section. 

Tissue Isolation from Mice

All studies were approved by the Institutional Care and Use Committee (IACUC), Children’s Hospital of Philadelphia under Protocol # IAC 17-000958.

We would like to thank the Reviewers and Editorial staff for their prompt and constructive evaluation of our work. We hope that our revised manuscript will be acceptable for publication. 

Sincerely, 

Paul Billings, Ph. D.

Division of Orthopaedic Surgery

The Children's Hospital of Philadelphia

Philadelphia, PA 19104

Ph: 267-426-9691/425-2077

Email: BillingsP@email.chop.edu

---

## [Editor Report · Decision Letter 1]

4 Feb 2020

Human and mouse activin genes: divergent expression of activin A protein variants

and identification of a novel heparan sulfate-binding domain in activin B

PONE-D-20-00545R1

Dear Dr. Billings,

We are pleased to inform you that your manuscript has been judged scientifically suitable for publication and will be formally accepted for publication once it complies with all outstanding technical requirements.

With kind regards,

Xin Zhang, Ph.D.

Academic Editor

PLOS ONE
---

## [Editor Report · Acceptance letter]

11 Feb 2020

PONE-D-20-00545R1 

Human and mouse activin genes: divergent expression of activin A protein variants
and identification of a novel heparan sulfate-binding domain in activin B 

Dear Dr. Billings:

I am pleased to inform you that your manuscript has been deemed suitable for publication in PLOS ONE. Congratulations! Your manuscript is now with our production department. 

With kind regards,

on behalf of

Dr. Xin Zhang 

Academic Editor

PLOS ONE